# Replacing Sedentary Time with Physical Activity and Sleep: A 24-Hour Movement Behaviour Perspective on Appetite Control

**DOI:** 10.3390/nu17193163

**Published:** 2025-10-07

**Authors:** Sundus Malaikah, Arwa Alruwaili, James P. Sanders, Alice E. Thackray, David J. Stensel, David Thivel, Joseph Henson, Alex V. Rowlands, Scott A. Willis, James A. King

**Affiliations:** 1Department of Clinical Nutrition, Faculty of Applied Medical Sciences, King Abdulaziz University, Jeddah P.O. Box 80215, Saudi Arabia; smalaikah@kau.edu.sa; 2Department of Respiratory Therapy, College of Applied Medical Sciences, King Saud bin Abdulaziz University for Health Sciences, Riyadh P.O. Box 11481, Saudi Arabia; ruwailiar@ksau-hs.edu.sa; 3King Abdullah International Medical Research Center, Riyadh P.O. Box 11481, Saudi Arabia; 4National Centre for Sport and Exercise Medicine, School of Sport, Exercise and Health Sciences, Loughborough University, Loughborough LE11 3TU, UK; j.sanders2@lboro.ac.uk (J.P.S.); a.e.thackray@lboro.ac.uk (A.E.T.); d.j.stensel@lboro.ac.uk (D.J.S.); s.willis2@lboro.ac.uk (S.A.W.); 5Centre for Lifestyle Medicine and Behaviour, Loughborough University, Loughborough LE11 3TU, UK; 6NIHR Leicester Biomedical Research Centre, University Hospitals of Leicester NHS Trust and the University of Leicester, Leicester LE5 4PW, UK; jjh18@leicester.ac.uk (J.H.); alex.rowlands@leicester.ac.uk (A.V.R.); 7Faculty of Sport Sciences, Waseda University, Tokyo 169-8050, Japan; 8Department of Sports Science and Physical Education, The Chinese University of Hong Kong, Shatin, New Territories, Hong Kong 999077, China; 9Clermont Auvergne University, EA3533 Laboratory of the Metabolic Adaptations to Exercise Under Physiological and Pathological Conditions (AME2P), CRNH, 63000 Clermont-Ferrand, France; david.thivel@uca.fr; 10Diabetes Research Centre, College of Life Sciences, University of Leicester, Leicester LE5 4PW, UK

**Keywords:** appetite, energy balance, eating behaviour, physical activity, sedentary time, food reward

## Abstract

**Background:** Physical activity, sedentary behaviour, and sleep are interdependent components of the 24 h movement profile that may influence appetite control. While acute exercise can alter appetite perceptions and food reward, less is known about how reallocating time between daily behaviours affects appetite outcomes under free-living conditions. **Methods:** We applied isotemporal-substitution modelling in a cross-sectional study of 130 young, healthy, active adults. Accelerometer-derived estimates of sedentary time, light physical activity (LPA), moderate-to-vigorous physical activity (MVPA), and sleep were analysed in relation to energy intake (food diaries, laboratory meals), subjective appetite perceptions, appetite-related hormones (acylated ghrelin, PYY, leptin), and psychological traits, including food reward (Leeds Food Preference Questionnaire, LFPQ), food cravings (Control of Eating Questionnaire, CoEQ), and eating behaviour traits (Three-Factor Eating Questionnaire, TFEQ). **Results:** Reallocating 30 min/day of sedentary time to MVPA was associated with higher energy intake in free-living (+113 kcal/day, 95% CI: 34–192) and laboratory settings (+120 kcal/day, 95% CI: 55–185), along with greater postprandial hunger and prospective food consumption, reduced fullness, elevated fasting acylated ghrelin, and lower postprandial PYY. No associations were observed for reallocations to LPA or sleep. Furthermore, sedentary time reallocations were unrelated to leptin or psychological eating traits assessed by the LFPQ, CoEQ, or TFEQ. **Conclusions:** In this population, reallocating sedentary time to MVPA was linked to physiological and behavioural compensation consistent with elevated energy demands, whereas reallocating to LPA or sleep showed no associations. Trait-level eating behaviours were unaffected, suggesting MVPA influences appetite primarily through acute physiological rather than enduring cognitive or hedonic pathways.

## 1. Introduction

Appetite control remains a key focus of scientific enquiry given the persistent global burden of overweight and obesity [1]. Recent advances in the neurobiology of appetite have underpinned the development of new pharmacotherapies for obesity, many of which target central appetite-regulatory pathways [2]. The clinical success of these agents [3,4] underscores the pivotal role of appetite in eating behaviour and body weight regulation. However, pharmacological treatments alone cannot fully address the scale of the obesity crisis. Non-pharmacological strategies that modulate appetite remain essential, both as standalone approaches for individuals not using medication and as complementary tools alongside pharmacotherapy [5,6]. A deeper understanding of how such interventions influence appetite—across both its homeostatic and hedonic dimensions—is therefore a priority.

One promising avenue is the interaction between movement behaviours and appetite control. Sedentary behaviour has long been associated with dysregulated appetite and weight gain [7], a relationship confirmed with modern device-based assessment methods [8]. In contrast, regular physical activity appears to enhance appetite sensitivity, enabling more accurate compensation for prior energy intake [9,10]. This “fine-tuning” effect is characterised by greater fasting hunger coupled with stronger postprandial satiety [11], aligning intake more closely with energy demands. Mechanistic studies suggest that exercise-induced changes in appetite-related peptides (e.g., ghrelin, glucagon-like peptide-1 [GLP-1], peptide YY [PYY]) may contribute to this response [12,13], while behavioural evidence indicates that physical activity can also modulate hedonic aspects of eating—reducing the reward value of high-fat foods and improving control over food cravings [14]. Sleep also plays a critical role in appetite regulation, with insufficient or disrupted sleep linked to increased hunger, altered satiety signalling, and greater energy intake [15,16,17,18]. When considered alongside physical activity and sedentary behaviour, sleep represents an additional behaviour that may shape appetite regulation within the context of the full 24 h day. Together, these findings position daily movement behaviours—including physical activity, sedentary time, and sleep—as potentially powerful non-pharmacological regulators of appetite.

Yet much of the existing evidence derives from tightly controlled laboratory studies or small-scale trials that may not capture the complexity of appetite regulation in free-living populations [19,20], often focusing on one behaviour alone. Addressing this gap requires methods that account for how daily movement behaviours interact within the constraints of a 24 h day. Isotemporal substitution analysis offers one such approach [21]. By modelling the effects of reallocating time between sedentary behaviour, physical activity, and sleep, it provides ecologically valid estimates of how these behaviours influence health outcomes [22,23,24]. When applied to appetite regulation, isotemporal substitution allows for a more integrated understanding of how movement behaviours collectively shape both homeostatic and hedonic components of appetite.

Building on this rationale, the present study used a deeply-phenotyped adult cohort to examine how replacing device-measured sedentary time with different physical activity intensities and sleep influences energy intake and appetite control. Appetite and related hormonal responses were assessed dynamically during a mixed-meal tolerance test, complemented by validated measures of food craving and reward. These analyses provide novel evidence on the relationship between movement behaviours and appetite regulation in a real-world context, with implications for the design of non-pharmacological strategies to support weight management.

## 2. Materials and Methods

### 2.1. Ethical Approval and Participants

A convenience sample of 130 participants were recruited (including men and women) from the local community. Participants were aged 18–55 years with a Body Mass Index (BMI) between 16.5 and 35.0 kg/m^2^. All participants were healthy, weight stable (<3 kg change in the 12 weeks prior), not dieting (or using medications that affect body weight), and had no history of metabolic or cardiovascular disorders. Female participants self-reported not being pregnant. Smokers were included (smoking status was included as a co-variate in statistical models) but we excluded people who vape or use e-cigarettes as the impact on study outcomes is less clear. Written informed consent was provided by all, and the study procedures adhered to the principles of the Declaration of Helsinki [25]. This study was approved by the Loughborough University Ethics Review Sub-Committee (project ID: 10980; 30 September 2022).

### 2.2. Study Design

This cross-sectional study involved two laboratory visits: one for familiarisation and a second for primary assessments. Visits were separated by at least one week, during which free-living physical activity, sedentary behaviour, sleep, and free-living energy intake were monitored. Participants were instructed to fast overnight (12 h) and abstain from caffeine, alcohol, and structured exercise for 24 h before each visit. With the acquired cross-sectional data, isotemporal substitution modelling was employed to examine the associations between reallocating time from sedentary behaviour to light physical activity (LPA), moderate-to-vigorous physical activity (MVPA), or sleep and multiple appetite and energy intake-related outcomes, while accounting for the finite nature of the 24 h day.

#### 2.2.1. Visit 1: Eligibility and Familiarisation

Participants attended the laboratory to confirm eligibility and undergo familiarisation with all study procedures. Demographic information was collected and assessments for anthropometry and resting metabolic rate (RMR) were completed. Food cravings during the previous week were assessed using the Control of Eating Questionnaire (CoEQ) [26].

Height and body mass were measured using a wireless measuring station (Seca Ltd., Hamburg, Germany) to the nearest 0.1 cm and 0.1 kg, respectively, and BMI (body mass/height^2^) was calculated.

RMR was measured using breath-by-breath indirect calorimetry (Cortex Metalyzer 3B, Leipzig, Germany), following standard protocols [27]. Participants rested in a quiet, supine position for 30 min, during which expired air was continuously sampled. RMR was calculated using the Weir equation from the final 20 min of data, excluding the initial 10 min to eliminate non-steady-state data [28].

#### 2.2.2. 24 h Movement Behaviour Assessment

After Visit 1, participants wore a triaxial accelerometer (GENEActiv, ActivInsights Ltd., Kimbolton, UK) on their non-dominant wrist for 24 h per day over seven days to monitor levels of physical activity, sedentary time and sleep. Participants followed their usual routines and logged sleep/wake times and any device removal. Data, recorded at 100 Hz, were downloaded and processed using the GGIR R-package (http://cran.r-project.org. accessed on 1 August 2025) [29,30]. Dynamic acceleration (Euclidean Norm minus 1 g [ENMO]) was averaged over 5 s epochs and reported in milligravitational units (mg), alongside a sleep detection algorithm to estimate sleep duration [30,31]. ENMO reflects the average dynamic acceleration throughout the day (24 h), calculated as the vector magnitude of the three acceleration axes with 1 g subtracted and any negative values set to zero. This metric is automatically computed within the GGIR R-package [29,30] and provides an indicator of total physical activity over the 24 h period [22,30]. Data were excluded if post-calibration errors exceeded 0.01 g (10 mg), valid wear time was less than three days (defined as ≥16 h per day), or if data were missing for any 15 min interval across the 24 h cycle. Non-wear time was identified based on a 15 min window showing a standard deviation below 13 mg or a range less than 50 mg on at least two of the three axes, as previously described [22]. Missing data were imputed using average ENMO values from corresponding time points on other days [31,32]. This ensured that outcome variables covered the entire 24 h cycle (1440 min) for each participant. Sleep windows and sleep duration within the window were determined using the HDCZA automated algorithm, with sleep defined from initial sleep onset to the final awakening of the night [31,33]. For each exposure variable, the mean across all valid days was calculated.

The following sleep characteristics were obtained: sleep duration (total accumulated sleep duration within the sleep window discounting any wake time) and wake after sleep onset (WASO, the total number of minutes that a person is awake after having initially fallen asleep). Activity levels were categorized as sedentary (<40 mg), light (LPA, 40–<100 mg), or moderate-to-vigorous physical activity (MVPA; ≥100 mg) [34,35].

#### 2.2.3. Free-Living Energy Intake Assessment

Participants completed a three-day weighed food dairy—two weekdays and one weekend day—to assess habitual dietary intake. To minimise reporting errors, participants were given instructions, a food scale, and encouraged to record intake in real time. Records were reviewed and analysed by a trained researcher using Nutritics software version 6.13 (Nutritics Ltd., Dublin, Ireland) to estimate daily energy and macronutrient intake.

#### 2.2.4. Visit 2: Appetite and Energy Intake Assessment

Participants arrived at the laboratory at 08:00 after a 12 h overnight fast (Figure 1). On arrival, they completed the Leeds Food Preference Questionnaire (LFPQ). At ~08:15, participants began a mixed-meal tolerance test (MM-TT). Baseline appetite ratings (hunger, fullness, satisfaction, prospective food consumption) were recorded using visual analogue scales (VAS) [36], and a fasting venous blood sample was taken before consumption of a standardised meal (08:15–08:30). VAS ratings were then completed every 30 min for 240 min, with additional venous blood samples collected at 30, 60, and 120 min post-meal. At 13:00 (30 min after the MM-TT) and 17:00, participants consumed *ad libitum* lunch and dinner meals and were then provided with an evening snack bag to consume as desired until the next morning. Food intake from the snack bag was recorded the following day.

#### 2.2.5. Study Meals

In line with published guidance [38], the MM-TT meal (porridge, whole milk, honey) was calculated as 25% of estimated daily energy requirements (RMR × 1.4 × 0.25) and consisted of 55% carbohydrates, 15% protein, and 30% fat. Meal energy was determined by multiplying participants measured RMR by 1.4 (to reflect the sedentary study conditions) and subsequently by 0.25 (typical share of daily energy intake from breakfast in the UK). Ref. [39] *Ad libitum* meals included a standardised pasta lunch (72% CHO, 12% protein, 16% fat) and a dinner of curry, naan, and basmati rice (71% CHO, 8% protein, 21% fat). Energy intake was calculated from food consumed, with initial weights recorded and manufacturer values used for energy estimation. The snack bag contained commercially available items (e.g., cookies, cereal bars, mini rolls).

#### 2.2.6. Leeds Food Preference Questionnaire

The LFPQ is a validated computer-based tool for assessing food preference and reward [40,41]. It measures liking and wanting for 16 foods across four categories (high-/low-fat savory/sweet). Participants select their most wanted item from paired comparisons, and reaction times (adjusted for selection frequency) are used to compute implicit wanting. Explicit liking and wanting are rated using 100 mm VAS, with fat and sweet bias scores calculated from these responses.

#### 2.2.7. Blood Sampling and Biochemical Analyses

Venous blood samples were collected into pre-chilled EDTA monovettes (Sarstedt, Leicester, UK) and centrifuged at 2383× *g* for 10 min (4 °C). The resulting plasma was aliquoted into 2 mL cryovials and stored at −80 °C. Acylated ghrelin samples were processed separately to preserve the acyl group [42]. Hormone concentrations (acylated ghrelin, total PYY, leptin) were measured using enzyme-linked immunosorbent assays (ELISA) kits from commercial suppliers (Bioquote Ltd., York, UK; Merck Millipore, Darmstadt, Germany; and Bio-Techne Ltd., Abingdon, UK, respectively) with within-batch CVs < 5.14%. Acylated ghrelin and total PYY were measured at fasting and postprandial time points (0, 30, 60, and 120 min) while leptin was assessed at baseline only (fasted).

### 2.3. Statistical Analysis

All analyses were performed using SPSS version 28 (SPSS Inc., Chicago, IL, USA). Data distribution was assessed using Kolmogorov–Smirnov tests and histograms. Results are presented as mean ± SD (normal), median (IQR) (non-normal), or *n* (%) (categorical) as appropriate. Total area under the curve (AUC) for postprandial responses was calculated using the trapezoid rule.

Isotemporal substitution analysis using generalized linear models was employed to investigate the association of reallocating sedentary time to LPA, MVPA, or sleep duration (minus WASO) on appetite-related outcomes. Isotemporal substitution is a recommended tool in observational studies utilising measures based on time [21]. In this context, the “replacement” of sedentary behaviour by another behaviour represents a statistical simulation within the regression models: it estimates the expected change in outcomes if time spent sedentary were reallocated, rather than reflecting an actual experimental manipulation. To investigate these associations, all of the movement behaviours except for the one being displaced (in this case sedentary time) were concurrently entered into the regression model, as shown below:

Appetite-related outcome (y) = β0 + (β1) sleep duration + (β2) WASO + (β3) sedentary time + (β4) LPA + (β5) MVPA + (β6) covariates—with sedentary time (β3) eliminated from the model.

Importantly, incorporating sleep duration and WASO, along with other behaviours, ensures that the time reallocation is modeled and standardized to a specific time frame (such as 24 h or 1440 min). As a result, it is not necessary to include total duration as an additional covariate in the model. The regression coefficient for each behaviour in this model indicates the change in association that would occur if a unit of time spent in sedentary time is replaced with that behaviour.

We ran two regression models. Model 1 adjusted for age, sex, ethnicity, and smoking status (with postprandial AUC outcomes additionally adjusted for fasting values). Model 2 included the same covariates with the addition of BMI, given its potential role as an attenuator of associations between movement behaviours and appetite-related outcomes [43]. Although results from both models are reported, Model 2 is considered the primary model of interest because it accounts for adiposity as a key confounder. Data from Model 1 are reported in Appendix A. Multicollinearity was checked using both pairwise correlations among predictors and variance inflation factors (VIFs) from Model 2. All pairwise correlations were ≤0.444 and VIFs were ≤1.577, indicating no evidence of problematic multicollinearity.

Missing data (3.4% for fasting bloods; 1.7% for appetite and field-based assessment of energy intake) were imputed across 20 datasets using regression models that included age, sex, ethnicity, BMI, and smoking status as predictors.

Given the exploratory nature of this study, formal *a priori* power calculations were not conducted, and no adjustments were made for multiple comparisons. Consequently, the findings should be interpreted cautiously, recognizing the risk of type I error. Results are best considered in the context of broader patterns, using beta-coefficients (β), 95% confidence intervals (CIs), and exact *p*-values. These outcomes are intended to guide the planning of future hypothesis-driven experimental research.

To evaluate the adequacy of the final sample (*n* = 119), we conducted a post hoc power analysis aligned with our primary isotemporal substitution models (Model 2) where laboratory energy intake (outcome) was regressed on the movement behaviours (total predictors = 9, error *df* = 109). At a two-sided alpha of 0.05 and 80% power, this design could detect partial effects of approximately partial R^2^ ≈ 0.07 for a single movement behaviour. Using the observed standard error of the MVPA coefficient (derived from its 95% CIs), this corresponded to a minimal detectable difference of ~94 kcal·day^−1^ per 30 min reallocation of sedentary time to MVPA. The observed association (+120 kcal·day^−1^; 95% CIs: 55, 185) therefore exceeded this threshold and corresponded to a partial R^2^ ≈ 0.11, indicating an adequate sample size.

## 3. Results

### 3.1. Participant Characteristics

Out of the total sample (*n* = 130), 11 individuals were excluded due to missing 24 h movement behaviour data related to technical issues—leaving a final sample of 119 participants. Key characteristics of these individuals are described in Table 1. Participants’ appetite, appetite-related hormone, food reward and energy intake data are shown in Appendix A. Overall, men and women were fairly equally represented, with most participants identifying as White, Indian or Asian. Generally, the sample was young, physically active and exhibiting healthy body weight.

### 3.2. Energy Intake

Based on food diary records, reallocating 30 min of sedentary time to MVPA was associated with a 113 (34, 192) kcal·d^−1^ higher daily energy intake (*p* = 0.005) (Figure 2, Model 2). Reallocating time from sedentary time to LPA or sleep was unrelated to daily energy intake measured via food diaries. Based on laboratory measured energy intake, reallocating 30 min of sedentary time to MVPA was associated with a 120 (55, 185) kcal·d^−1^ higher daily energy intake (*p* < 0.001) (Figure 2, Model 2). Reallocating time from sedentary time to LPA or sleep was not associated with energy intake measured in the laboratory.

### 3.3. Appetite-Related Hormones

Reallocating 30 min of sedentary time to MVPA was associated with a 639 (−1143, −135) pg·mL·2 h^−1^ lower postprandial PYY AUC (*p* = 0.013) (Model 2, Table 2). Reallocating 30 min of sedentary time to MVPA was also associated with higher fasting acylated ghrelin concentrations (0.01 (0.00, 0.19) pg·mL^−1^) (*p* = 0.045) (Model 2, Figure 2). Reallocation of 30 min of sedentary time for LPA was associated with higher postprandial PYY concentrations (550 (35, 1065) pg·mL·2 h^−1^) (*p* = 0.036) (Model 2, Table 2).

### 3.4. Perceived Ratings of Appetite

Reallocating 30 min of sedentary time to MVPA was associated with higher postprandial PFC (4 h AUC: +545 mm; 95% CI: 93–996; *p* = 0.018; Model 2, Table 2). There was also a trend toward greater postprandial hunger (+481 mm; 95% CI: –49 to 1011; *p* = 0.075) and lower postprandial fullness (–454 mm; 95% CI: –937 to 30; *p* = 0.066) when reallocating sedentary time for MVPA. No other associations were observed for reallocations of sedentary time to physical activity (LPA or MVPA) or sleep, whether appetite perceptions measurements were fasted or postprandial.

### 3.5. Food Reward, Cravings and Dietary Eating Traits

Reallocation of 30 min of sedentary time to LPA, MVPA and sleep was unrelated to all measured outcomes regarding food reward (LFPQ), food cravings (CoEQ) and dietary eating traits (TFEQ) (Table 3).

## 4. Discussion

The present study used isotemporal substitution modelling to examine the impact of reallocating sedentary time to sleep, LPA, or MVPA on multiple aspects of appetite control in a predominantly young, healthy, and active cohort. We investigated outcomes spanning energy intake, subjective appetite perceptions, appetite-related hormones, and psychological eating behaviour traits. Reallocation of sedentary time to MVPA was consistently associated with higher energy intake, greater postprandial appetite, and hormonal changes indicative of increased energy demands (higher fasting acylated ghrelin, lower postprandial PYY). No associations were observed for reallocations of sedentary time to LPA or sleep, and there were no links with trait-level eating behaviours, such as dietary restraint, disinhibition, food cravings, or food reward. Taken together, these findings suggest that in young, healthy, and active adults, reallocating time to MVPA under free-living conditions may elicit physiological and behavioural compensation to maintain energy balance, potentially through enhanced sensitivity of short-term appetite signals. In contrast, reallocations to LPA and sleep do not appear to meaningfully influence appetite regulation or energy intake in this population.

A key finding was that reallocating sedentary time to MVPA was associated with higher daily energy intake, with each 30 min reallocation corresponding to an additional 113 kcal (free-living measured) to 120 kcal (laboratory measured) consumed. No such associations were detected for reallocations to LPA or sleep. This likely reflects the characteristics of our cohort, who were weight-stable, highly active, and therefore required greater energy intake to meet the increased demands of higher physical activity. Such a pattern is consistent with the “activity-stat” hypothesis, which proposes that active individuals finely adjust energy intake to meet expenditure. For example, previous studies have shown that active individuals consume more energy and protein than their less active peers [44], report greater hunger and reduced satiety following meals [45], and demonstrate enhanced appetite sensitivity by downregulating intake after high-energy preloads [46]. These findings collectively suggest that active individuals may possess a more responsive appetite control system that helps align energy intake with expenditure.

This interpretation is supported by our subjective appetite data. Reallocating sedentary time to MVPA was associated with greater postprandial hunger and prospective food consumption, and with lower fullness during the MM-TT. No associations were observed for fasting appetite ratings, likely reflecting the variability and limited sensitivity of single time-point VAS measures, whereas repeated postprandial ratings more reliably capture satiety dynamics [38]. That the associations emerged only in the postprandial period could suggest that MVPA may sharpen meal-related signalling and appetite cues when energy is most needed. The absence of associations for reallocations to sleep or LPA is plausibly explained by their relatively modest energetic contribution in this cohort, alongside the homogeneity and generally healthy sleep and LPA pattern of participants.

Further insight comes from the appetite-related hormone data. Few studies have explored the interaction between habitual physical activity and appetite peptides [47], though some evidence suggests that active individuals have higher fasting acylated ghrelin concentrations than inactive controls [48]. In line with this, reallocating sedentary time to MVPA in our cohort was associated with higher fasting acylated ghrelin concentrations, consistent with an adaptive signal to stimulate intake in response to elevated energy demands [49]. Lower postprandial PYY with greater MVPA extends this picture, suggesting that satiety signalling may be downregulated in active individuals to facilitate energy replenishment [50]. In contrast, acylated ghrelin responses to meals did not differ, which may be expected given the marked postprandial decline observed across individuals (limiting variability in data). Furthermore, leptin, which primarily reflects longer-term energy reserves and adiposity [51], was not influenced by movement behaviour reallocations, likely because participants were weight stable.

The consistency of MVPA–energy intake associations across both dietary records and laboratory-based assessments strengthens confidence in these findings. While self-reported diaries are prone to underestimation [52], this bias is broadly systematic, whereas laboratory-based assessments provide greater precision but lower ecological validity. The convergence of both methods therefore provides robust evidence that reallocating sedentary time to MVPA is associated with higher energy intake in this population. Notably, the confidence intervals for associations with energy intake were wide, particularly when intake was assessed via dietary records compared with laboratory measures. This likely reflects both the limited variability in movement behaviours within our cohort and the substantial inter-individual variability in dietary intake responses, highlighting the importance of replication in larger and more diverse samples.

We also examined hedonic and cognitive influences on eating behaviour. Using the LFPQ [42], we assessed food reward (implicit and explicit liking and wanting), alongside the CoEQ (craving control and food cravings) [26] and the TFEQ (restraint, disinhibition, hunger) [53]. In contrast to our findings for energy intake and appetite physiology, reallocating sedentary time to MVPA, LPA, or sleep was unrelated to any of these outcomes. Prior research has suggested that habitual MVPA is associated with lower liking and wanting for high-fat foods [54,55] and attenuated neural reward responses to food cues [56,57], though such effects are most evident in less active or higher-adiposity populations, and the evidence from intervention studies is mixed [58]. Our null findings may therefore reflect the characteristics of our sample, namely young, active, and healthy adults with a well-regulated appetitive system and limited variability in eating behaviour traits. Furthermore, the questionnaires employed primarily capture trait-like or longer-term dispositions, which may be relatively insensitive to the subtle habitual reallocations in movement and sleep assessed here. This pattern suggests that the influence of physical activity on appetite in this population is more likely mediated through acute physiological pathways as opposed to enduring cognitive or hedonic mechanisms.

Our overall findings have both practical and clinical relevance. In our young, active cohort, reallocating sedentary time to MVPA was linked to higher intake and a hunger-promoting profile, indicating compensatory eating rather than appetite suppression. Accordingly, physical activity guidance (particularly for MVPA) should be paired with targeted nutritional guidance to align intake with goals. In a weight-management context, this means coupling MVPA with strategies to manage overall energy intake to avoid unintended surplus, whereas in a performance context, it means proactive fueling to meet higher demands while maintaining nutrient quality. These implications are population-specific and non-causal, and should be confirmed in other populations.

This study has several strengths, including the precise accelerometer-based assessment of 24 h movement behaviours, the use of isotemporal substitution modelling to capture real-world behavioural trade-offs, and the integration of comprehensive outcome measures spanning energy intake, appetite perceptions, hormone responses, and psychological traits. To our knowledge, this is the first application of isotemporal substitution in this research area. The following limitations should also be noted: the cross-sectional design prevents causal inference and raises the possibility of reverse causality, the homogeneous sample restricts applicability to older, less active, or populations living with obesity, and the limited variability in sleep and LPA may have reduced power to detect associations. Moreover, participants were recruited through convenience sampling from the local community, which may introduce selection bias and limit the extent to which these findings can be extrapolated to broader populations. Additionally, given the exploratory nature of this work we did not adjust for multiple comparisons, and therefore individual associations should be interpreted with caution, with emphasis placed on the overall pattern of findings rather than isolated results. Outcomes were measured on a single occasion, which may not fully capture longer-term patterns or account for the considerable inter-individual variability in appetite regulation. Finally, despite some inherent correlation among 24 h movement behaviours, collinearity diagnostics showed no problematic multicollinearity (*r* ≤ 0.444, VIFs ≤ 1.577), though some imprecision is still possible.

## 5. Conclusions

In conclusion, we hypothesized that reallocating sedentary time to different 24 h movement behaviours would be linked to altered appetite regulation. Our findings support this for MVPA, but not LPA or sleep; reallocating sedentary time to MVPA was associated with higher energy intake, greater postprandial appetite, and hormonal changes consistent with compensatory responses to increased energy expenditure in young, healthy, and active adults. By contrast, trait-level eating behaviours were not associated with these reallocations, reinforcing that physical activity may influence appetite primarily through acute physiological rather than cognitive mechanisms. These findings provide new insight into the interplay between daily movement behaviours and appetite regulation, and highlight the importance of considering compensatory eating in the context of MVPA promotion. Future work should employ longitudinal and interventional designs and examine subgroups (e.g., by sex or BMI category) to test causality and assess generalizability across diverse populations.

## Figures and Tables

**Figure 1 nutrients-17-03163-f001:**
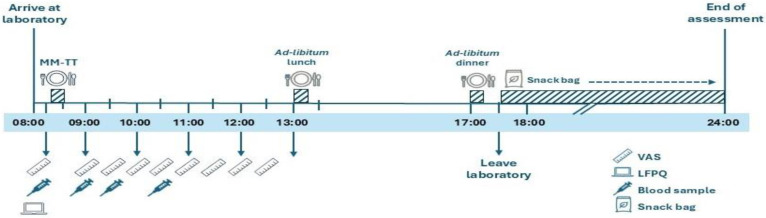
Schematic illustration of study visit 2. LFPQ, Leeds Food Preference Questionnaire; MM-TT, mixed-meal tolerance test; VAS, visual analogue scale. ‘Reproduced from Alruwaili et al., 2025, *Appetite*, Jun 11:108194 [37], under the terms of the Creative Commons Attribution License (CC BY 4.0). https://doi.org/10.1016/j.appet.2025.108194.

**Figure 2 nutrients-17-03163-f002:**
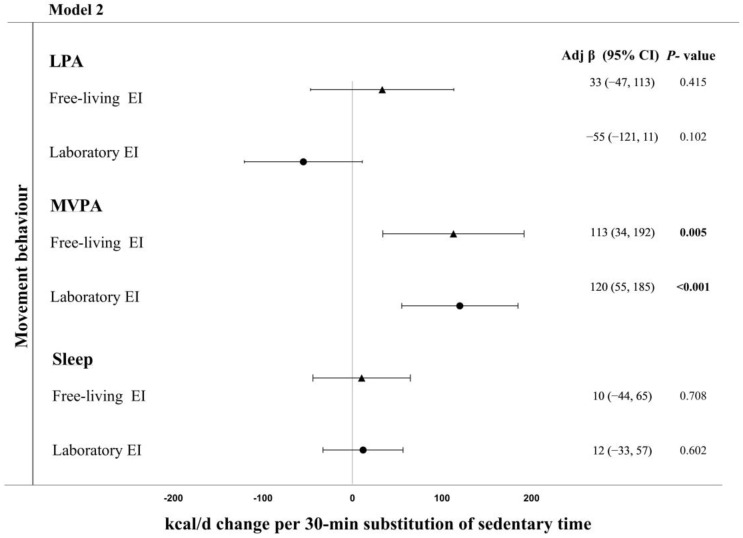
Adjusted change in energy intake (measured in the laboratory and field) associated with replacing 30 min of sedentary time with sleep or physical activity in Model 2 (adjusted for age, sex, ethnicity, smoking status, and BMI). Bold indicates significant.

**Table 1 nutrients-17-03163-t001:** Participant characteristics (*n* = 119).

Variable	Mean ± SD/Median (IQR)/*n* [%]
Sex (male)	70 [58.8%]
Ethnicity	
White	53 [44.5%]
Indian	28 [23.5%]
Asian	25 [21.0%]
Mixed	6 [5.0%]
Black	3 [2.5%]
Arab	2 [1.7%]
Latino	2 [1.7%]
Age (years)	24 (9)
Body mass (kg)	70.5 ± 12.8
BMI (kg·m^−2^)	23.6 (5.1)
**Movement behaviours**	
Sedentary time (min·d^−1^)	704 ± 103 Range: 409 to 1088
LPA (min·d^−1^)	194 ± 53 Range: 56 to 336
MVPA (min·d^−1^)	112 ± 50 Range: 25 to 273
Sleep time during sleep period (min·d^−1^)	336 ± 76 Range: 106 to 477
WASO during sleep period (min·d^−1^)	95 ± 49 Range: 26 to 327
Total sleep period (min·d^−1^)	430 ± 68 Range: 220 to 595

Data are presented for *n* = 119. Values are mean ± standard deviation (SD) for normally distributed variables, median and interquartile range (IQR) for non-normally distributed variables, and frequencies and percentages for categorical variables. LPA, light physical activity; MVPA, moderate-vigorous physical activity; WASO, wakefulness after sleep onset.

**Table 2 nutrients-17-03163-t002:** Adjusted change in appetite-related hormones, and perceived ratings of appetite associated with replacing 30 min of sedentary time with sleep or physical activity in Model 2 (adjusted for age, sex, ethnicity, smoking status, and BMI).

	LPA		MVPA		Sleep	
	β-Coefficient (95% CI)	*p*-Value	β-Coefficient (95% CI)	*p*-Value	β-Coefficient (95% CI)	*p*-Value
**Appetite-related hormones**						
Fasting leptin (pg·mL^−1^) *	0.03 (−0.1, 0.1)	0.652	−0.06 (−0.15, 0.03)	0.111	−0.06 (−0.12, 0.00)	0.106
Fasting PYY (pg·mL^−1^)	4.6 (−1.8, 11.0)	0.162	−0.8 (−7.1, 5.6)	0.815	−0.2 (−4.6, 4.1)	0.916
PYY AUC (2 h, pg·mL^−1^)	**549.8 (34.8, 1064.9)**	**0.036**	**−639.3 (−1143.2, −135.3)**	**0.013**	52.7 (−294.5 to 399.8)	0.766
Fasting acylated ghrelin (pg·mL^−1^) *	0.06 (−0.04, 0.14)	0.234	**0.10 (0.00, 0.19)**	**0.045**	0.00 (−0.04, 0.06)	0.648
Acylated ghrelin AUC (2 h, pg·mL^−1^) *	−0.03 (−0.06, 0.03)	0.514	0.03 (−0.03, 0.09)	0.566	−0.03 (−0.06, 0.03)	0.359
**Perceived ratings of appetite**						
Fasting fullness (mm)	−0.33 (−3.00, 2.33)	0.809	1.00 (−1.65, 3.66)	0.456	0.09 (−1.74, 1.93)	0.927
Fullness AUC (4 h, mm)	6.5 (−482.5, 495.4)	0.979	−453.8 (−936.9, 29.5)	0.066	−286.7 (−618.8 to 45.5)	0.091
Fasting hunger (mm)	−0.2 (−3.4, 3.0)	0.915	0.8 (−2.4, 4.0)	0.635	0.0 (−2.2, 2.2)	0.990
Hunger AUC (4 h, mm)	−70.6 (−607.4, 466.3)	0.797	480.8 (−49.1, 1010.7)	0.075	299.3 (−65.5 to 662.0)	0.108
Fasting PFC (mm)	0.30 (−1.86, 2.42)	0.789	1.77 (−0.34, 3.85)	0.101	0.53 (−0.93, 1.99)	0.467
PFC AUC (4 h, mm)	−168.9 (−621.5, 283.8)	0.464	**544.6 (93.4, 995.8)**	**0.018**	175.8 (−132.2 to 483.8)	0.263
Fasting satisfaction (mm)	1.84 (−1.08, 4.71)	0.217	0.39 (−2.46, 3.24)	0.785	0.72 (−1.23, 2.70)	0.465
Satisfaction AUC (4 h, mm)	−138.7 (−624.9, 347.7)	0.576	−135.2 (−611.9, 341.5)	0.578	−192.5 (−521.6 to 136.5)	0.251

Each coefficient reflects the predicted change in the specified appetite-related outcome associated with reallocating 30 min of sedentary time to the specified behaviour (LPA, MVPA, sleep), holding total time constant. Values are presented as adjusted means with 95% CI for *n* = 119. Data were analysed using generalised linear models with a normal distribution and identity link function or gamma distribution with a log link function indicated by *. Model 2 adjusted for age, sex, smoking status, ethnicity and BMI. AU, arbitrary units; CI, confidence interval; LPA, light physical activity; MVPA, moderate-to-vigorous physical activity. * Values represent coefficients on the log-scale for models with gamma distribution and log link function. Bold indicates significant.

**Table 3 nutrients-17-03163-t003:** Adjusted change in food reward (LFPQ), food cravings (CoEQ) and eating traits (TFEQ) associated with replacing 30 min of sedentary time with sleep or physical activity in Model 2 (adjusted for age, sex, ethnicity, smoking status, and BMI).

	LPA	MVPA		Sleep Duration		
	β-Coefficient (95% CI)	*p*-Value	β-Coefficient (95% CI)	*p*-Value	β-Coefficient (95% CI)	*p*-Value
**TEFQ**						
Cognitive restraint (0–21)	0.00 (−0.50, 0.51)	0.957	−0.03 (−0.50, 0.44)	0.901	−0.12 (−0.46, 0.25)	0.521
Disinhibition (0–16)	−0.15 (−0.45, 0.18)	0.365	0.12 (−0.18, 0.45)	0.418	−0.10 (−0.30, 0.15)	0.476
Hunger (0–14)	0.16 (−0.25, 0.55)	0.482	0.30 (−0.06, 0.70)	0.104	0.00 (−0.27, 0.25)	0.967
Total score	−0.03 (−0.72, 0.66)	0.943	0.42 (−0.30, 1.15)	0.247	−0.21 (−0.66, 0.25)	0.397
**CoEQ**						
Craving control (AU)	−3.93 (−12.00, 4.14)	0.34	2.26 (−5.70, 10.22)	0.577	3.27 (−2.22, 8.77)	0.242
Craving for Sweet (AU)	2.03 (−7.90, 11.96)	0.687	−2.82 (−12.63, 6.98)	0.575	6.18 (−0.60, 13.00)	0.074
Craving for Savoury (AU)	2.3 (−7.2, 11.7)	0.638	2.3 (−7.1, 11.6)	0.630	−3.4 (−10.0, 3.1)	0.302
Positive Mood (AU)	6.3 (−13.5, 26.1)	0.107	−6.3 (−22.5, 0.9)	0.102	3.4 (−1.8, 8.7)	0.201
**LFPQ**						
Fasting fat explicit liking (mm)	−0.5 (−2.0, 1.0)	0.527	0.1 (−1.5, 1.7)	0.907	−0.3 (−1.4, 0.8)	0.618
Fasting fat explicit wanting (mm)	−0.3 (−2.0, 1.4)	0.754	0.6 (−1.2, 2.3)	0.470	0.0 (−1.1, 1.1)	0.992
Fasting fat implicit wanting (AU)	−1.0 (−4.2, 2.2)	0.536	−0.1 (−3.3, 3.0)	0.930	−0.3 (−2.5, 1.9)	0.789
Fasting fat relative preference (AU)	−0.36 (−1.56, 0.84)	0.538	−0.18 (−1.42, 1.05)	0.777	0.18 (−0.63, 0.89)	0.658
Fasting taste explicit liking (mm)	−0.96 (−3.21, 1.30)	0.401	−0.75 (−2.94, 1.44)	0.504	−0.03 (−1.53, 1.48)	0.984
Fasting taste explicit wanting (mm)	−1.05 (−3.24, 1.14)	0.355	−0.30 (−2.50, 1.89)	0.777	0.45 (−1.02, 1.93)	0.548
Fasting taste implicit wanting (AU)	0.57 (−4.09, 5.23)	0.806	−3.51 (−8.07, 1.05)	0.134	1.05 (−2.03, 4.14)	0.512
Fasting taste relative preference (AU)	−0.00 (−6.10, 5.30)	0.891	−2.20 (−7.80, 3.40)	0.442	0.60 (−3.20, 4.50)	0.745

Each coefficient reflects the predicted change in the specified appetite-related outcome associated with reallocating 30 min of sedentary time to the specified behaviour (LPA, MVPA, sleep), holding total time constant. Values are presented as adjusted means with 95% CI for *n* = 119. Data were analysed using generalised linear models with a normal distribution and identity link function. Model 2 adjusted for age, sex, smoking status, ethnicity, and BMI. AU, arbitrary units; CI, confidence interval; CoEQ, Control of Eating Questionnaire; ES, effect size; LFPQ, Leeds Food Preference Questionnaire; LPA, light physical activity; MVPA, moderate-to-vigorous physical activity; TFEQ, Three-Factor Eating Questionnaire.

## Data Availability

The data presented in this study (anonymized) are available on request from the corresponding author.

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
