# Peer review of "Replacing Sedentary Time with Physical Activity and Sleep: A 24-Hour Movement Behaviour Perspective on Appetite Control"

_nutrients, 2025, doi:10.3390/nu17193163_

Round 1

Reviewer 1 Report

Comments and Suggestions for Authors

Dear corresponding Author, thank you for submiting your work to Nutrients journal and congratulations for the research.

Brief summary: Cross-sectional study on 119 active adults examining effects of reallocating sedentary time towards physical activity and sleep on appetite control using isotemporal substitution models.

General comments: The cross-sectional design limits causal inference and the posibility of reverse causality. The homogenous population (young, active, healthy) reduces generalizability. Statistical power calculations and corrections for multiple comparisons are missing. The exploratory nature of the study is appropriatley acknowledged but some results might reflect type I errors.

Specific comments:

  • Lines 35-41: The main results should specify that they refer only to reallocation towards MVPA, not towards LPA or sleep. Try to reflect on this, it appeared strange to me...
  • Line 242: The lack of corrections for multiple comparisons needs a more thorough discussion on the risks of false positives and could give misunderstandings.
  • Table 1: Consider adding information on inter-individual variability of movement behaviors. Also it doesn't have correct formating because the description of the second column is completly missing and doesn't make reading and comprehension immediate.
  • Figure 1: too small, I believe the template allows you to extend the image width.
  • Figure 2: The confidence intervals for associations with LPA and sleep are wide - discuss the implications for statistical power and explain better, it's not clear.
  • Line 387: The statement "This study has several strengths..." is self-referential and extremely overvalued since the limitations are decidedly more evident but much less valorized.
  • Lines 392-394: The limitation regarding single measurement of outcomes is important but could be expanded considering intra-individual variability of appetite.

Ultimately the work in this form is not yet ready for publication.

Reviewer 2 Report

Comments and Suggestions for Authors

Overall, I found this to be a solid and scientifically rigorous manuscript. The study is methodologically coherent, well written, and based on an appropriate analytical strategy. The Introduction is comprehensive and well justified, the Methods are clearly structured, and the Discussion integrates the findings with existing literature in a meaningful way. The results are presented with precision, and the overall narrative is consistent and scientifically sound.

That said, there are a few areas where the manuscript could be further strengthened. My detailed comments are provided below and focus on clarifying methodological aspects, improving the explanation of statistical modelling, highlighting significant results more explicitly, and enhancing the depth of the Discussion.

  • Comment: Materials and Methods – Sampling Strategy. The manuscript indicates that participants were recruited from the local community, which corresponds to a convenience sampling approach. This needs to be explicitly stated, as it limits the representativeness of the sample and restricts the generalizability of the findings to broader populations differing in age, health status, or physical activity levels.

  • Comment: Materials and Methods – Sample Size. The authors acknowledge that no a priori sample size calculation or power analysis was conducted. This is a notable methodological limitation, as the absence of a formal estimation weakens the justification for the chosen sample. I suggest that the authors include an ad hoc sample size or power calculation aligned with their statistical analysis plan. This would provide greater confidence in the adequacy of the study design. Given the final sample size (n = 119), it is likely that the study was sufficiently powered, but presenting such a calculation would strengthen the methodological rigor and the interpretation of the findings.

  • Comment: Study Design and Statistical Analysis. The description of isotemporal substitution modelling is not sufficiently clear. As currently presented, it may lead readers to believe that participants actually replaced sedentary time with other behaviors, whereas this is a statistical modelling approach rather than an experimental manipulation. I strongly recommend that the authors provide a clearer explanation of the isotemporal substitution method. Specifically, it would be useful to briefly introduce the concept already in the Study Design section, and then expand on it more technically in the Statistical Analysis section. This should explicitly clarify that the “replacement” of sedentary behavior by light physical activity, moderate-to-vigorous physical activity, or sleep represents a statistical simulation within the regression models. For readers and researchers who are not familiar with this analytical strategy, the current description is difficult to follow. Even for those with expertise in research methodology and statistics, the central idea of the study is not immediately clear without further clarification.

  • Comment: Results – Tables 2 and 3. The Results section presents extensive data in Tables 2 and 3, but the narrative text does not sufficiently highlight or explain the key findings, particularly those reaching statistical significance. As a reader, it is difficult to identify the most relevant outcomes without carefully examining the tables. I recommend that the authors provide a clearer description of the significant associations observed, especially regarding the substitution of sedentary time with MVPA and its effects on energy intake, appetite ratings, and appetite-related hormones.

  • Comment: Discussion. The Discussion is coherent with the study’s findings and provides a reasonable integration with existing literature. However, several aspects could be improved. First, the section tends to reiterate results rather than offering deeper interpretation. I recommend reducing redundancy and expanding the critical discussion of the mechanisms and implications. Second, the practical and clinical relevance of the findings (e.g., for physical activity guidelines or appetite regulation strategies) should be discussed in greater depth. Third, some limitations are acknowledged (e.g., cross-sectional design, absence of power calculation), but others are insufficiently addressed. In particular, the use of convenience sampling and the potential impact of multicollinearity on the regression models deserve explicit recognition. Finally, the Discussion would benefit from a clearer return to the initial hypothesis and a stronger conclusion that synthesizes the key message, alongside more specific future research directions (e.g., longitudinal interventions, subgroup analyses by sex or BMI). These additions would substantially strengthen the interpretability and impact of the manuscript.

Round 2

Reviewer 1 Report

Comments and Suggestions for Authors

I have thoroughly reviewed the revisions made by the authors and consider the manuscript, in its current form, ready for publication.

Reviewer 2 Report

Comments and Suggestions for Authors

The authors have adequately addressed most of the reviewer’s comments. They clarified the sampling strategy as convenience sampling, added a post hoc power analysis to strengthen methodological rigor, and provided clearer explanations of isotemporal substitution modelling. The Discussion was revised to reduce redundancy, expand on mechanisms and implications, and highlight practical and clinical relevance. Limitations, including potential multicollinearity, were tested and transparently acknowledged.